# The role of Rif1 in telomere length regulation is separable from its role in origin firing

**Calla B Shubin[1,2], Carol W Greider[1]\***

[1]Department of Molecular Biology and Genetics, Johns Hopkins University School of Medicine, Baltimore, United States; [2]Biochemistry, Cellular and Molecular Biology Graduate Program, Johns Hopkins University School of Medicine, Baltimore, United States

**Abstract** To examine the established link between DNA replication and telomere length, we tested whether firing of telomeric origins would cause telomere lengthening. We found that *RIF1* mutants that block Protein Phosphatase 1 (PP1) binding activated telomeric origins but did not elongate telomeres. In a second approach, we found overexpression of ΔN-Dbf4 and Cdc7 increased DDK activity and activated telomeric origins, yet telomere length was unchanged. We tested a third mechanism to activate origins using the *sld3-A mcm5-bob1* mutant that de-regulates the pre-replication complex, and again saw no change in telomere length. Finally, we tested whether mutations in *RIF1* that cause telomere elongation would affect origin firing. We found that neither *rif1-Δ1322* nor *rif1$_{HOOK}$* affected firing of telomeric origins. We conclude that telomeric origin firing does not cause telomere elongation, and the role of Rif1 in regulating origin firing is separable from its role in regulating telomere length.

**\*For correspondence:**
cgreider@jhmi.edu

**Competing interests:** The authors declare that no competing interests exist.

## Introduction

Telomeres are critical components of chromosome function. They consist of tandem repeats of simple G- and T-rich sequences at the ends of eukaryotic chromosomes. Telomeres shorten with repeated rounds of DNA replication, and this shortening is counterbalanced by the enzyme telomerase (*Greider and Blackburn, 1985*). Telomere lengths differ between species, but, in all healthy cells, length is maintained within a defined length distribution (*Greider, 1996*). How this distribution of telomere lengths is established and maintained is not well understood. Yet the maintenance of the length distribution is crucial, as short telomeres signal a DNA damage response that limits cell division (*d'Adda di Fagagna et al., 2003*; *Enomoto et al., 2002*; *Hemann et al., 2001*; *IJpma and Greider, 2003*). In human patients with defective telomere maintenance, short telomere syndromes manifest as degenerative diseases, including bone marrow failure and pulmonary fibrosis (*Armanios and Blackburn, 2012*). Conversely, inappropriate telomere lengthening can contribute to immortalization of cancer cells (*Autexier and Greider, 1996*; *Greider, 1990*) and to a predisposition to cancer (*McNally et al., 2019*). Understanding the molecular basis of telomere length maintenance thus has important implications for human disease.

Evidence from a number of sources suggests that establishment of telomere length equilibrium is linked to DNA replication. Telomere elongation in yeast only occurs after passage of a replication fork (*Dionne and Wellinger, 1998*). Mutations in components of the replication fork alter telomere length in yeast; for example, components of lagging strand synthesis such as DNA polymerase α, RFC, Dna2, and Fen1, cause telomere elongation (reviewed in *Greider, 2016*). Further, DNA polymerases α and δ and DNA primase are all essential for de novo telomere addition by telomerase (*Diede and Gottschling, 1999*). The canonical yeast single-strand telomere end-binding complex,

Cdc13/Stn1/Ten1, is considered a telomere-specific Replication Protein A (t-RPA) (*Gao et al., 2007*). t-RPA interacts directly with DNA polymerase α and has been proposed to assist DNA replication through telomere repeats, which further links telomere length and DNA replication (*Gao et al., 2010*). Recent evidence indicating that the telomere binding protein Rif1 regulates telomeric origin firing suggested origin firing may regulate telomere elongation (*Greider, 2016*).

Rif1 was first identified for its role in regulating telomere length. In *Saccharomyces cerevisiae*, deletion of *RIF1* leads to extensive telomere elongation (*Hardy et al., 1992*). This role of Rif1 in regulating telomere length is conserved in distantly related yeast such as *Schizosaccharomyces pombe* (*Kanoh and Ishikawa, 2001*) and *Candida glabrata* (*Castaño et al., 2005*). In *S. cerevisiae*, Rif1 binds to telomeres through its interaction with the telomere-specific double-stranded DNA binding protein Rap1. Evidence that Rif1 regulates telomeric origin firing (*Cornacchia et al., 2012*; *Davé et al., 2014*; *Hayano et al., 2012*; *Peace et al., 2014*; *Yamazaki et al., 2012*) helped explain the longstanding observation that telomeric origins fire late in S phase. Transplanting an origin sequence, which is known to fire early at its endogenous locus, to a telomere, caused that origin to instead fire late (*Ferguson et al., 1991*; *Ferguson and Fangman, 1992*; *Wellinger et al., 1993*). Further, cell synchronization showed that telomeres incorporate BrdU late in S phase (*Raghuraman et al., 2001*). These studies proposed that telomeric origins are 'late origins' or 'dormant origins', terms used to characterize telomeric origins that don't fire at all in a given cell cycle. Rif1 binding to telomeres and repressing nearby origin firing (*Davé et al., 2014*; *Hiraga et al., 2014*; *Mattarocci et al., 2014*; *Peace et al., 2014*) presented a nice explanation for the late replication of telomeric origins.

Several groups showed that the mechanism of Rif1 repression of origin firing works through its interaction with Protein Phosphatase 1 (PP1, *GLC7* in *S. cerevisiae*). PP1 can dephosphorylate key substrates in the pre-replication complex (Pre-RC) including Mcm4 (*Davé et al., 2014*; *Hiraga et al., 2014*; *Mattarocci et al., 2014*) and thus block origin activation. DDK phosphorylation of Mcm4 is required for origin activation at the initiation of S phase, so PP1 antagonizes DDK phosphorylation to block origin firing. Rif1 recruitment of PP1 to repress origin firing is highly conserved across species including *S. pombe*, *Drosophila*, and mammalian cells, although the PP1 interaction motifs, RVxF and SILK, are located in a different region of the protein in mammalian cells (*Sreesankar et al., 2012*; *Xu et al., 2010*). This conservation of function indicates repressing origin firing is a core function of Rif1.

The conserved functions of Rif1 in both telomere length regulation and in repression of telomeric origin firing suggested the attractive hypothesis that origin firing may be directly linked to telomere length regulation (*Greider, 2016*). To critically test this idea, we genetically manipulated origin firing and measured the effects on telomere length. We looked at several mutants known to affect origin firing: first, a *RIF1* mutant, *rif1-pp1bs,* which cannot recruit PP1, and therefore cannot repress origin firing; second, ΔN-dbf4 Cdc7 overexpression, a cell cycle stabilized DDK that increases DDK activity; finally, *sld3-A mcm5-bob1*, a double mutant that can bypass both the Rad53 inactivation of Sld3 (*Deegan et al., 2016*; *Zegerman and Diffley, 2007*) and the DDK phosphorylation of Mcm5 (*Hardy et al., 1997*) to activate origins. In all three cases, telomeric origin firing was increased, and telomere length did not change. Finally, we looked at two *RIF1* mutants, *rif1-Δ1322* and *rif1*$_{HOOK}$, which have long telomeres, and found that they did not change origin firing. Therefore, we conclude that the Rif1 function in telomere length regulation is separate from its function in PP1 recruitment and regulation of origin firing.

## Results

Deletion of *RIF1* activates telomeric origin firing (*Davé et al., 2014*; *Hafner et al., 2018*; *Mattarocci et al., 2014*; *Peace et al., 2014*). This origin activation requires Rif1 binding to PP1 through canonical binding motifs RVxF and SILK, (*Davé et al., 2014*; *Hiraga et al., 2014*; *Mattarocci et al., 2014*). To examine whether origin firing regulates telomere length, we made mutations in the RVxF and SILK motifs to generate a Rif1-PP1 binding site mutant, termed *rif1-pp1bs*. To validate that PP1 binding was disrupted, we examined Mcm4 phosphorylation by western blot. Others have shown that phosphorylation of Mcm4 is increased in *rif1Δ* and in *rif1*-PP1 binding site mutants (*Davé et al., 2014*; *Hiraga et al., 2014*; *Mattarocci et al., 2014*). We found *rif1Δ* and our *rif1-pp1bs* mutant showed increased levels of Mcm4 phosphorylation compared to *WT* in both

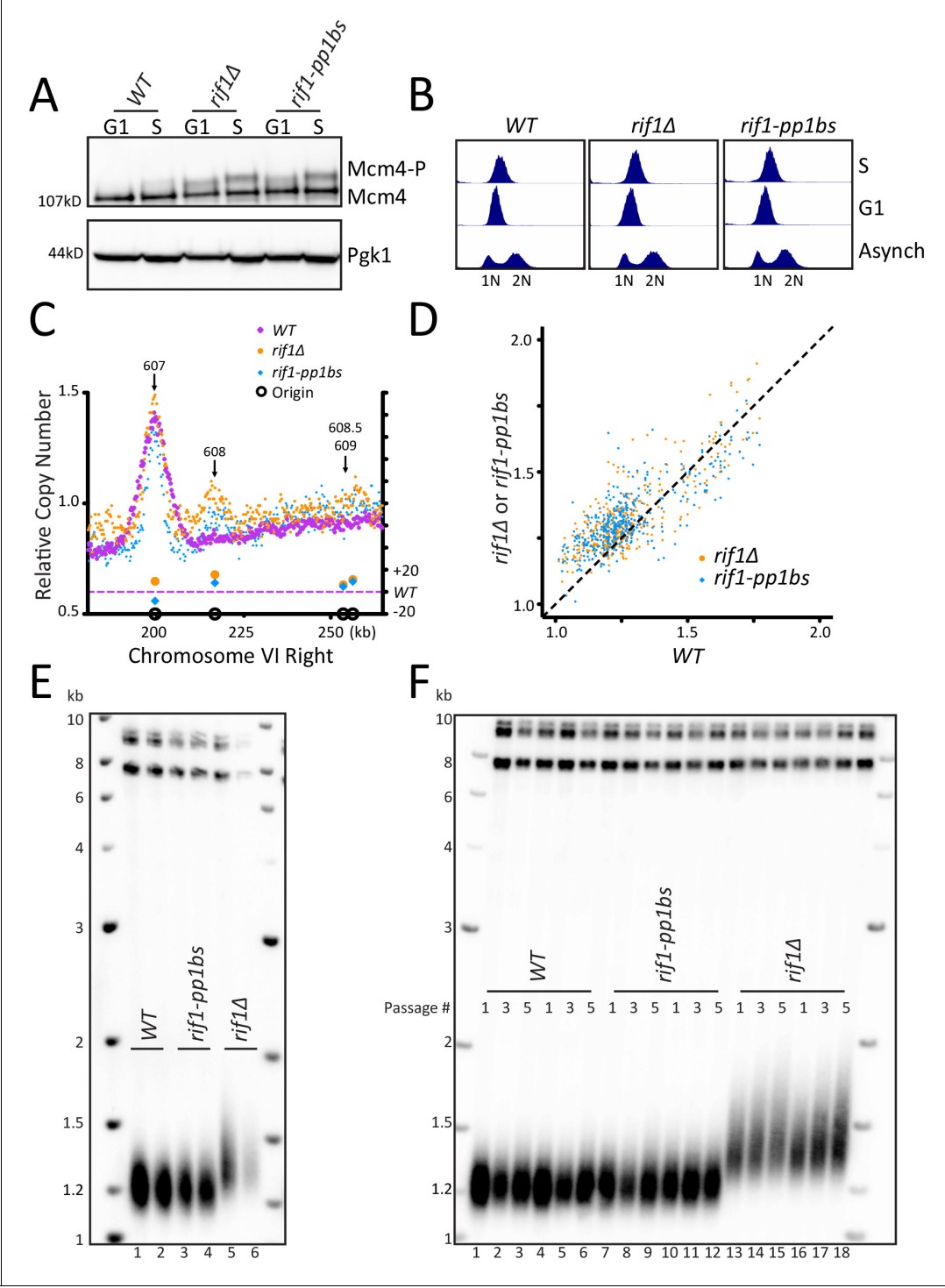

**Figure 1.** Disruption of Rif1 binding to PP1 activates origin firing but does not increase telomere length. (A) *WT, rif1Δ*, or *rif1-pp1bs* cells were arrested in G1 with alpha factor (G1) and then released into HU (S), and the level of Mcm4 phosphorylation was detected by western blot. Pgk1 is used as a loading control. (B) Cell cycle analysis of asynchronous, G1, and S phase arrested cells was measured using flow cytometry to follow DNA content. (C) The relative copy number of DNA sequences in S phase compared to G1 was plotted for the right arm of Chromosome VI. The X axis shows position across the chromosome, and normalized sequence read number is plotted on the left Y axis. The relative fraction of origins fired in *rif1Δ* or *rif1-pp1bs* is plotted on the right Y axis. *WT* is in purple, *rif1Δ* is in orange and *rif1-pp1bs* is in blue. Confirmed origins (from OriDB) are denoted on the X axis as open black circles. (D) The relative copy number for each of the confirmed OriDB origins in WT cells is plotted on the X axis and the relative copy number in *rif1Δ* (orange) or *rif1-pp1bs* (blue) is plotted on the Y axis. The relative copy number for each origin is scaled by 1.25 to account for expected

*Figure 1 continued on next page*

*Figure 1 continued*

copy number change during early S phase. (E) Southern blot showing telomere length of indicated strains. (F) Southern blot of indicated strains passaged in liquid culture for five passages.

The online version of this article includes the following figure supplement(s) for figure 1:

**Figure supplement 1.** Disruption of Rif1 binding to PP1 activates origin firing across the genome.
**Figure supplement 2.** *rif1-pp1bs* gives similar results to the mutation *rif1-R/S* (*Mattarocci et al., 2014*).

G1 and S phase, as other groups have shown (*Figure 1A* and *Figure 1—figure supplement 1C*). Lambda phosphatase treatment confirmed that the upper band on the western is due to Mcm4 phosphorylation (*Figure 1—figure supplement 1A,B*). Having confirmed *rif1Δ* and our *rif1-pp1bs* increase Mcm4 phosphorylation, next we examined their effects on both origin firing and telomere length.

## Disruption of Rif1 binding to PP1 activates origin firing

To examine the effect of *rif1Δ* and *rif1-pp1bs* directly on origin firing, we analyzed DNA sequence copy number in S phase by whole genome sequencing, in a manner similar to *Müller et al., 2014*. We synchronized cells in G1 using alpha factor and collected a sample, then released cells into Hydroxyurea (HU) to arrest cells in early S phase and collected a second sample (*Figure 1B*, see Materials and methods). We used Illumina whole genome paired-end sequencing to determine genome sequence copy number for both the S phase and the G1 samples (see Materials and methods). We determined origins had fired when the ratio of sequence reads S/G1 was increased in a 1 kb window around the midpoint of the confirmed origins in OriDB (*Siow et al., 2012*). As a control, we compared three independent *WT* sequencing runs and determined a similar extent of origin firing occurred in each pairwise comparison of all OriDB origins (*Figure 1—figure supplement 1D*). We focused our attention on Chromosome VI-right telomeric origins ARS608, ARS608.5, and ARS609. *rif1Δ* and *rif1-pp1bs* samples showed increased telomeric origin firing compared to *WT* at the dormant telomeric origin ARS608 and to a lesser extent at ARS608.5 and ARS609 as expected from previous work. In contrast, the nearby early origin ARS607 fired efficiently in all strains (*Figure 1C*). Analysis across the entire Chromosome VI showed that the left telomeric origins also showed slight increase in activation in *rif1Δ* and *rif1-pp1bs* (*Figure 1—figure supplement 1E*). To determine the extent of activation of each origin, we used a method similar to *Hafner et al., 2018* and compared the relative copy number at each origin in *rif1Δ* and *rif1-pp1bs* to *WT*. As expected, there was higher relative copy number at most origins in the *rif1Δ* and *rif1-pp1bs* mutants, with the majority of the change seen in typically late/inefficient origins, demonstrated by a low relative copy number in *WT* cells (*Figure 1D*). To quantify the increase in origin firing, we calculated the difference in relative copy number of origins for *rif1Δ* and *rif1-pp1bs* to determine the relative fraction of origins fired compared to *WT* (see Materials and methods) and found both *rif1Δ* and *rif1-pp1bs* were statistically significantly increased compared to *WT* (*Figure 1—figure supplement 1F*). To visualize the extent of origin activation, we generated heatmaps representing sequence read counts for a 10 kb region centered at each confirmed OriDB origin. This showed an increase in origin firing in *rif1Δ* and *rif1-pp1bs* compared to *WT* (*Figure 1—figure supplement 1G*). Genomic locations where an increased ratio was seen, shown in orange at the center of the heatmap, mapped precisely to the midpoint of the confirmed origins. These results support the previous work (*Davé et al., 2014*; *Hiraga et al., 2014*; *Mattarocci et al., 2014*), which showed that disruption of Rif1 interaction with PP1 leads to origin activation.

## Telomeric origin activation does not increase telomere length

Having established that our *rif1-pp1bs* mutant activates telomeric origins, we asked whether telomeric origin firing increases telomere length. Southern blots of *rif1Δ* and *rif1-pp1bs* showed that, while *rif1Δ* had long telomeres as expected, *rif1-pp1bs* mutants showed telomere length similar to *WT* (*Figure 1E*). We passaged the *rif1-pp1bs* mutants in liquid culture to overcome any phenotypic delay in telomere lengthening and still found no change in telomere length compared to *WT*, whereas *rif1Δ* continued to elongate (*Figure 1F*). This result differs from a previous study, which found that telomeres were longer in a different *RIF1* mutant that disrupted PP1 binding

(*Kedziora et al., 2018*). To further probe this discrepancy, we examined additional *RIF1* PP1 binding mutants reported in the literature (*Figure 1—figure supplement 2A*). We generated the previously published *RIF1* PP1 binding mutant called *rif1-R/S* mutant from *Mattarocci et al., 2014*, and found it had wildtype telomere length similar to our *rif1-pp1bs* (*Figure 1—figure supplement 2A,B*). Next, we examined protein stability by western blot and found that that our mutant, *rif1-pp1bs*, and the *rif1-R/S* were stably expressed (*Figure 1—figure supplement 2C*). We also found that Mcm4 phosphorylation levels increased in *rif1-pp1bs* and *rif1-R/S* to a similar extent (*Figure 1—figure supplement 2D,E*). In contrast, the *RIF1* PP1 binding mutant reported in *Hiraga et al., 2014* and further analyzed in *Kedziora et al., 2018*, *rif1-pp1bsD*, in which two additional putative PP1 binding sites 3 and 4 were mutated, was not stably expressed (*Figure 1—figure supplement 2A,F*). To further examine the potential stability issues, we mutated the additional sites 3 and 4 in our construct to generate *rif1-pp1bs-4* and found this protein was also not stably expressed (*Figure 1—figure supplement 2A,G*). This data suggests that mutation of the additional two sites in *RIF1* may disrupt protein folding and cause degradation of Rif1 protein. Therefore, the long telomere phenotype seen in Kedziora et al., (and our unpublished data) can be attributed to the absence of Rif1 protein (*Kedziora et al., 2018*). We conclude, from our *rif1-pp1bs* and the Mattarocci et al. *rif1-R/S* mutant, that mutating the Rif1-PP1 binding site increases telomeric origin firing to a similar extent as a *rif1Δ*, but this increased firing does not result in telomere elongation.

## Increasing origin activation by DDK overexpression does not increase telomere length

Because of the different conclusion in the literature about *RIF1* mutants that disrupt PP1 binding, we sought an alternative approach to examine whether increased telomeric origin firing affects telomere length. Since PP1 dephosphorylates Mcm4, we wanted to increase the level of Mcm4 phosphorylation by kinase overexpression. This phosphorylation is carried out by the Dbf4-dependent regulatory kinase (DDK), which requires both a catalytic subunit, Cdc7, and a regulatory subunit, Dbf4 (*Sheu and Stillman, 2006*). To further increase DDK activity, we used ΔN-dbf4, a cell cycle stabilized form of Dbf4 with an N-terminal 65 amino acid truncation (*Ferreira et al., 2000*; *Sullivan et al., 2008*). We generated strains overexpressing Cdc7 and either ΔN-dbf4 or Dbf4 under the constitutive ADH1 promoter (Materials and methods). Western blots showed that ΔN-dbf4 levels were stabilized in G1 compared to the levels of full length Dbf4, which is degraded in G1, as previously shown (*Figure 2A*, *Figure 2—figure supplement 1A*). Mcm4 phosphorylation was increased in cells overexpressing ΔN-dbf4, or both ΔN-dbf4 and Cdc7 (*Figure 2B*, and *Figure 2—figure supplement 1A*). Cells overexpressing ΔN-dbf4 alone or in combination with Cdc7 showed a robust increase in telomeric ARS608 firing and to a lesser extent ARS608.5 and ARS609 compared to *WT* (*Figure 2C*, *Figure 2—figure supplement 1B*). Analysis of the entire Chromosome VI showed increased firing at both telomeric and non-telomeric origins with ΔN-dbf4 and Cdc7 overexpression (*Figure 2—figure supplement 1C*). Analysis of relative copy number at each origin showed overexpression of ΔN-dbf4 and also ΔN-dbf4 and Cdc7 together resulted in higher relative copy number at most origins compared to WT cells, with a dramatic increase in late firing origins (*Figure 2D*). Overexpression of both ΔN-dbf4 and ΔN-dbf4 and Cdc7 together showed a significant increase in global origin firing when compared to WT, further depicted by heatmap representation of relative copy number spanning 10 kb around the each origin (*Figure 2—figure supplement 1D,E*). Despite the strong activation of telomeric origins, and many other origins, in cells overexpressing ΔN-dbf4 and ΔN-dbf4 with Cdc7, we saw no change in telomere length (*Figure 2E*). We conclude that increasing origin activation through DDK overexpression increases origin firing but does not affect telomere length.

## Increasing origin activation using mutations in the pre-RC does not increase telomere length

As a third way to examine telomere length and origin firing, we tested known mutations in the pre-replication complex (pre-RC) that activate late/dormant origins. The Diffley lab showed that the double mutant *sld3-A mcm5-bob1* activates late origins (*Zegerman and Diffley, 2007*; *Zegerman and Diffley, 2010*). Mcm4 phosphorylation levels increased in *sld3-A bob1* double mutants (*Figure 3A*, *Figure 3—figure supplement 1A*). Analysis of origin firing showed a robust increase in origin firing in *sld3-A bob1* compared to *WT* for the dormant, telomeric origins ARS608, ARS608.5 and ARS609

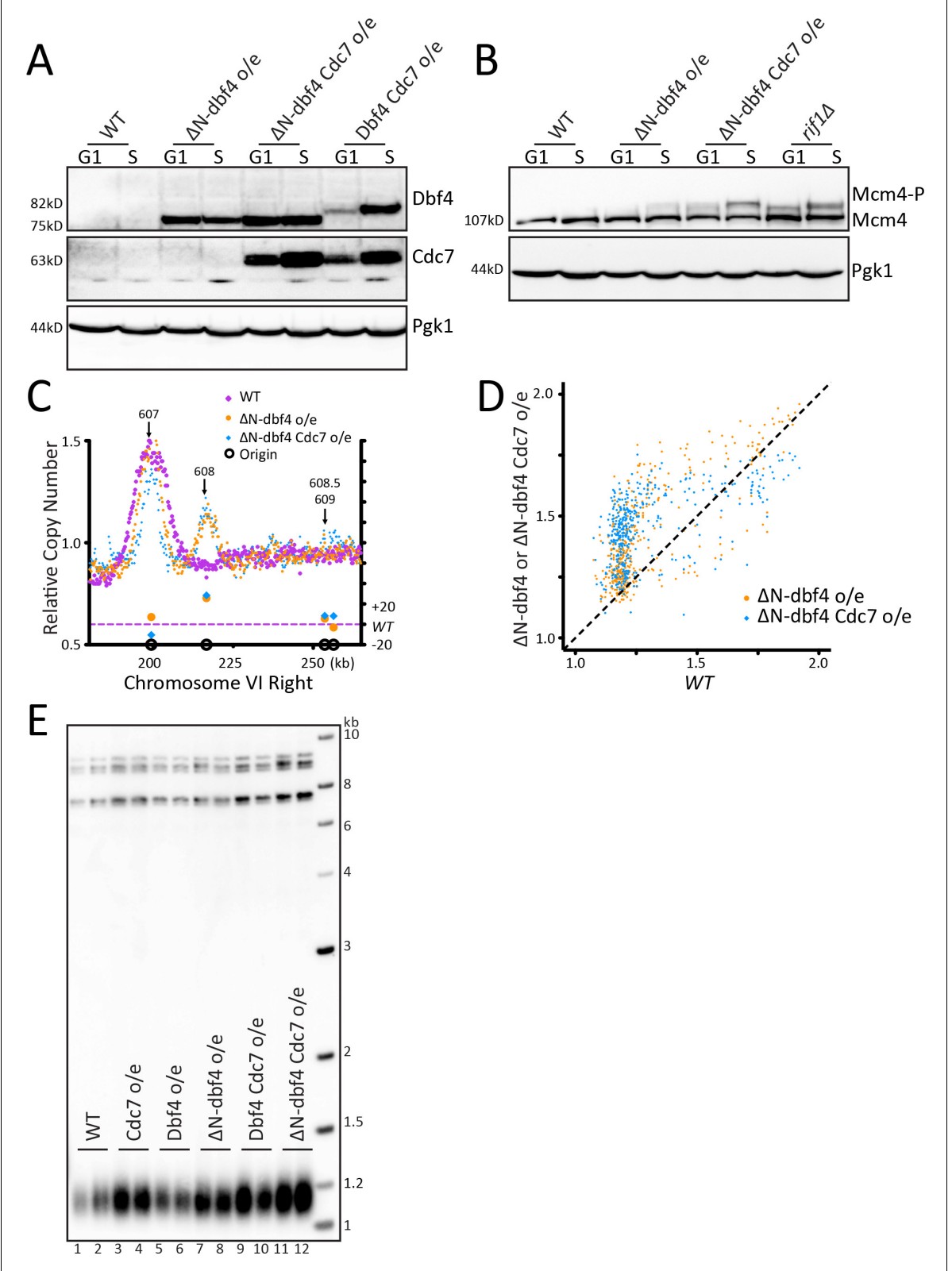

**Figure 2.** Increasing origin activation by DDK overexpression does not increase telomere length. (**A**) Western blot detecting the level of Dbf4 and Cdc7 in *WT* cells and cells overexpressing ΔN-dbf4, both ΔN-dbf4 and Cdc7, or both Dbf4 and Cdc7 (DDK) arrested in G1 or S. (**B**) The samples shown in A were probed on a western blot to detect Mcm4 phosphorylation levels. (**C**) The relative copy number of DNA sequences in S phase compared to G1 was plotted for the right arm of Chromosome VI, as in *Figure 1C*. WT is in purple, ΔN-dbf4 o/e is in orange and ΔN-dbf4 Cdc7 o/e is in blue. (**D**) For
*Figure 2 continued on next page*

*Figure 2 continued*

each of the confirmed OriDB origins, relative copy number for that given origin in WT cells is plotted on the X axis and the relative copy number in ΔN-dbf4 o/e (orange) or ΔN-dbf4 Cdc7 o/e (blue) is plotted on the Y axis, as in *Figure 1D*. (E) Southern blot analysis of telomere length of strains indicated.

The online version of this article includes the following figure supplement(s) for figure 2:

**Figure supplement 1.** DDK overexpression increases origin activation across the genome.

(*Figure 3B*, *Figure 3—figure supplement 1B* relating to *Figure 3*). Examination of all of Chromosome VI showed that most origins increased firing in *sld3-A bob1* compared to *WT* (*Figure 3—figure supplement 1C*). The peaks around each origin were narrower in *sld3-A bob1* than in *WT*; this may be due to depletion of the nucleotide pools in HU, decreasing replication fork progression when so many origins are activated at once, as noted in *Mantiero et al., 2011*. For example, in ARS607, the peak height is the same in *sld3-A bob1* compared to *WT* (*Figure 3B*), indicating a similar number of cells in which the origin fired; however, the peak width is narrower, indicating the fork was not able to travel as far in HU. Quantitation of the relative copy number at origins in *sld3-A bob1* compared to *WT* showed a significant increase in origin firing in *sld3-A bob1* (*Figure 3C*). Comparison of relative copy number at each origin showed early firing of almost all origins, which is particularly striking at the late *WT* origins (*Figure 3D*). Finally, heatmap analysis also confirmed early firing of almost all origins (*Figure 3—figure supplement 1D*). Remarkably, despite the robust telomeric origin activation, the *sld3-A bob1* double mutant maintained *WT* telomere length (*Figure 3E*). To examine this further, we passaged *WT* and *sld3-A bob1* cells in liquid culture for 6 days, and telomere length was maintained at *WT* length (*Figure 3F*). We conclude that increased origin firing in the *sld3-A bob1* double mutant does not lead to a change in telomere length. Taken together, our experiments using three different ways to activate origin firing indicate that the role of Rif1 in regulating origin firing is separable from its role in regulating telomere length.

## Mutations in Rif1 that increase telomere length do not affect origin firing

To take a different approach to critically examine any link in telomere length and origin firing, we looked at two Rif1 mutants that have long telomeres, *rif1-Δ1322* and *rif1_HOOK* (*Mattarocci et al., 2017*; *Mattarocci et al., 2014*; *Shi et al., 2013*). *rif1-Δ1322* is a C- terminal truncation that removes the Rap1 binding motif. Loss of Rap1 binding causes a substantial decrease in Rif1 telomeric localization (*Hafner et al., 2018*; *Hiraga et al., 2018*). We confirmed by western that the rif1-Δ1322 and the rif1_HOOK proteins were stably expressed (*Figure 4A*). *rif1-Δ1322* had long telomeres, although not as long as *rif1Δ* (*Figure 4B*) as noted by *Shi et al., 2013*, and *rif1_HOOK* showed long telomeres similar to *rif1Δ* as previously described (*Mattarocci et al., 2017*). We next examined whether this long telomere phenotype correlated with altered origin firing. We found no increase in Mcm4 phosphorylation levels in either *rif1-Δ1322* or *rif1_HOOK* (*Figure 4C* and *Figure 4—figure supplement 1A*). This lack of increased Mcm4 phosphorylation suggests that rif1-Δ1322 can still recruit PP1 to dephosphorylate Mcm4 even though it cannot localize to the telomere. Telomeric origins ARS608, ARS608.5, and ARS609 were not activated in either *rif1-Δ1322 or rif1_HOOK* mutants, and there was very little change in most origins along the entire Chromosome VI (*Figure 4D*, *Figure 4—figure supplement 1B,C*). Analysis of relative copy number at each origin compared to *WT* showed no change in origin firing in the *rif1_HOOK* mutant, while *rif1-Δ1322* showed an apparent small decrease in origin firing (*Figure 4E*) as seen previously (*Hafner et al., 2018*). Quantification of the relative copy number showed no significant increase in origin firing in *rif1-Δ1322* or *rif1_HOOK* compared to *WT* (*Figure 4—figure supplement 1D*). Heatmap representation of the data further demonstrates that neither *rif1-Δ1322* nor *rif1_HOOK* increase origin firing compared to *WT* (*Figure 4—figure supplement 1E*). These data suggest that a substantial increase in telomere length in *rif1_HOOK* mutant does not alter telomeric origin firing.

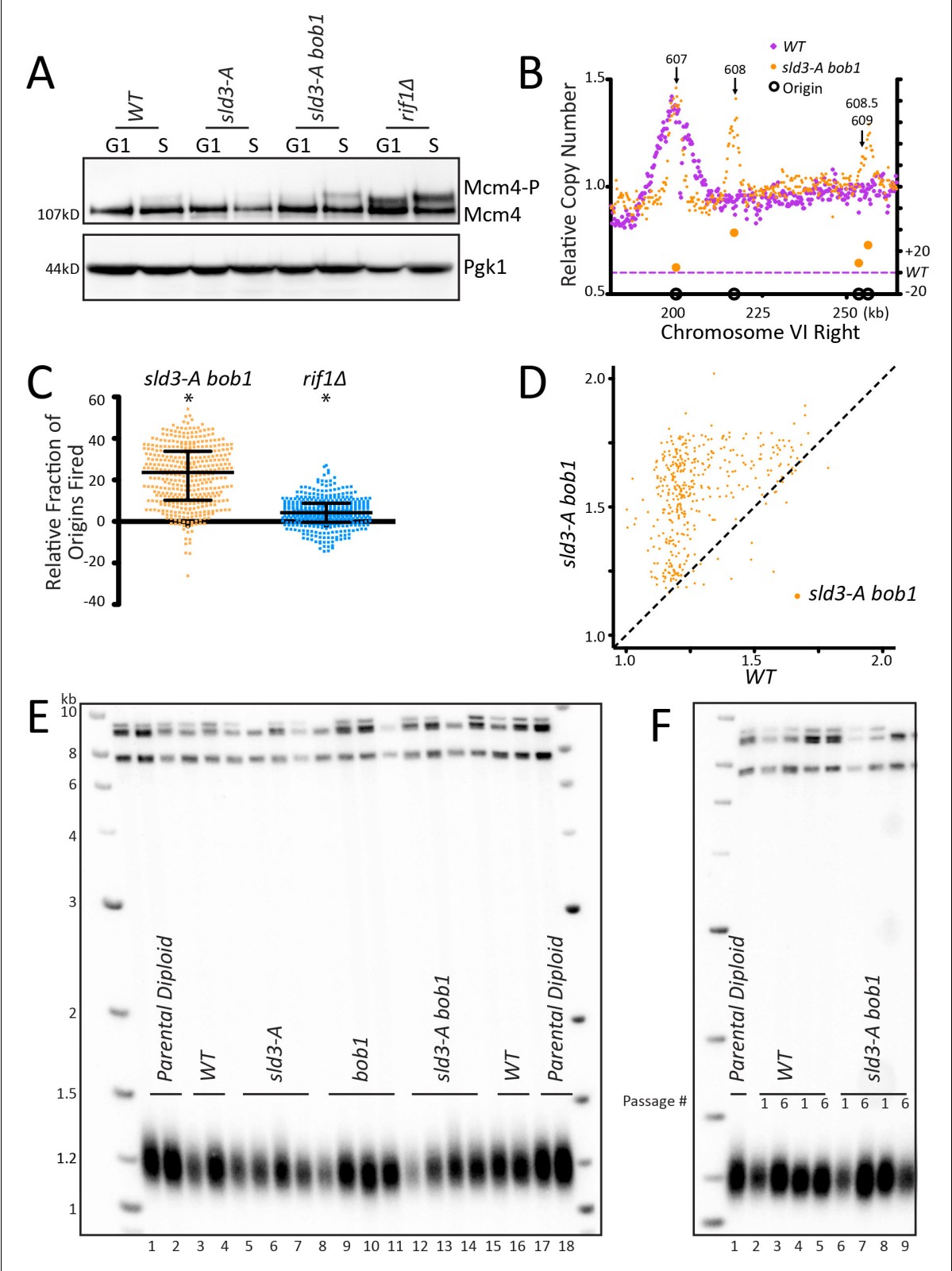

**Figure 3.** Increasing origin activation using mutations in the pre-RC does not increase telomere length. (**A**) *WT*, *sld3-A*, *sld3-A bob1*, and *rif1Δ* cells were arrested with alpha factor (G1) and then released into HU (S), and the level of phosphorylation of Mcm4 was detected by western. (**B**) The relative copy number of DNA sequences in S phase compared to G1 was plotted for the right arm of Chromosome VI as indicated in *Figure 1C*. *WT* is in purple and *sld3-A bob1* is in orange. (**C**) The relative copy number for 1 kb around the midpoint of each origin was calculated for each strain. For each

*Figure 3 continued on next page*

*Figure 3 continued*

peak, the *WT* value was subtracted from mutant value and multiplied by 100. A positive value indicates more origins fired in the population in the mutant compared to WT. Median and interquartile range are plotted over the distribution, and * indicates a significant difference by one-sided Wilcoxon signed rank test: *sld3-A bob1* (p<0.0001) and *rif1Δ* (p<0.0001). (D) For each of the confirmed OriDB origins, relative copy number for that given origin in WT cells is plotted on the X axis and in *sld3-A bob1* (orange) is plotted on the Y axis, as in *Figure 1D*. (E) Southern blot analysis of telomere length of strains indicated. (F) Southern blot of strains passaged in liquid culture for six passages.

The online version of this article includes the following figure supplement(s) for figure 3:

**Figure supplement 1.** Mutations in the pre-RC increase origin activation across the genome.

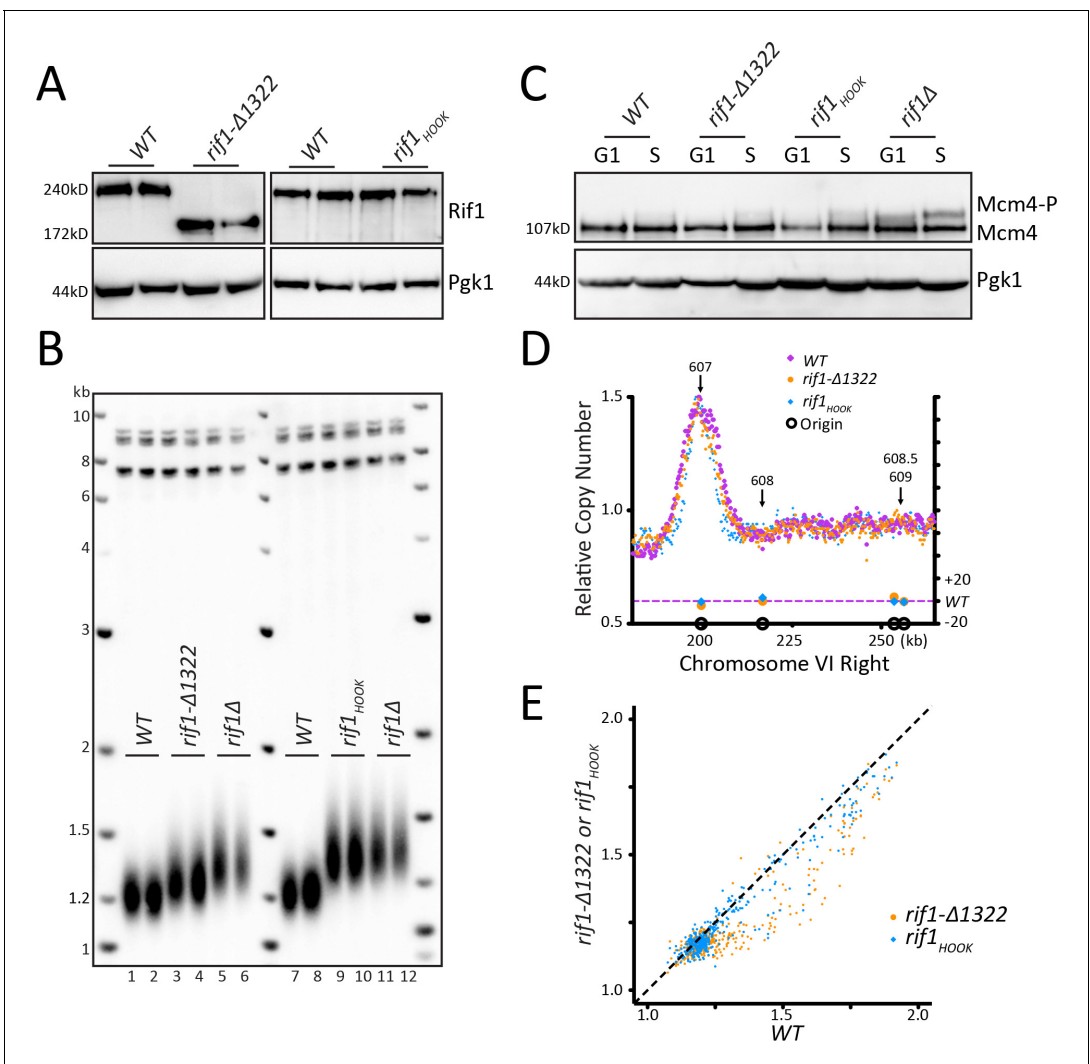

**Figure 4.** Mutations in Rif1 that increase telomere length do not affect origin firing. (A) Rif1 protein levels measured by western blot for *WT*, *rif1-Δ1322*, and *rif1$_{HOOK}$*. (B) Southern blot analysis of telomere length of strains indicated. (C) *WT*, *rif1-Δ1322*, *rif1$_{HOOK}$*, and *rif1Δ* cells were arrested with alpha factor (G1) and then released into HU (S), and the level of phosphorylation of Mcm4 was detected by western. (D) The relative copy number of DNA sequences in S phase compared to G1 was plotted for the right arm of Chromosome VI, as in *Figure 1C*. *WT* is in purple, *rif1-Δ1322* is in orange, and *rif1$_{HOOK}$* is in blue. (E) For each of the confirmed OriDB origins, relative copy number for that given origin in *WT* cells is plotted on the X axis and the copy number in *rif1-Δ1322* (orange) or *rif1$_{HOOK}$* (blue) is plotted on the Y axis, as in *Figure 1D*.

The online version of this article includes the following figure supplement(s) for figure 4:

**Figure supplement 1.** Mutations in Rif1 that increase telomere length do not affect origin firing across the genome.

## Discussion

Telomere length regulation has been linked to DNA replication in many studies over the last 25 years. The finding that Rif1, a regulator of telomere length, also regulates replication origin firing (*Davé et al., 2014*; *Hayano et al., 2012*; *Mattarocci et al., 2014*; *Peace et al., 2014*; *Yamazaki et al., 2012*) suggested that origin firing may directly affect telomere elongation (*Dionne and Wellinger, 1998*; *Greider, 2016*). We have shown, using three unique genetic pathways: Rif1 mutations, DDK overexpression, and bypass mutants in the pre-RC, that activating telomeric origins is not sufficient to increase telomere length. While a very subtle effect on telomere length due to origin activation might not easily be resolved by a Southern blot, we predict that, if origin firing were linked to telomere length, then greater telomeric origin activation would lead to greater telomere elongation. To compare the increase in origin firing among the mutants, we computed a relative activation score for ARS608. The ratio of S phase activation of ARS608 relative to ARS607 in the mutants were normalized to that of the WT control to generate the relative ARS608 activation score (see Materials and methods). We found that *sld3-A bob1* and ΔN-dbf4 Cdc7 overexpression showed the highest relative activation scores, 24.36 and 25.83, respectively. ΔN-dbf4 overexpression alone had a score of 14.09, while *rif1-pp1bs* was 10.15 and *rif1Δ* was 6.54. In contrast, *rif1-Δ1322* and *rif1_{HOOK}* had low ARS608 scores of 1.50 and 2.33 respectively, but have long telomeres. Since telomere elongation is significant in *rif1Δ* and this mutant has the lowest relative activation score of those that increase telomeric origin firing, we expect that the mutants with higher scores would show significantly long telomeres if origin firing is linked to telomere elongation. Instead they all showed *WT* length telomeres. Our data indicate that the two well established functions of Rif1, regulating origin firing and regulating telomere length, represent independent, separable functions of Rif1.

Our results differ from those of Kedziora et al., who conclude that PP1 binding to Rif1 regulates both origin firing and telomere length (*Kedziora et al., 2018*). This group made mutations in four regions of Rif1 predicted to bind PP1, while our study and the study by Mattarocci mutated only two sites (*Mattarocci et al., 2014*). To address the discrepancy in our conclusions, we mutated the additional sites 3 and 4 as described by *Kedziora et al., 2018*, and found that this 4 site mutant *rif1-pp1bs-4* showed very low protein levels on a western blot (*Figure 1—figure supplement 2*) suggesting the additional site 3 and 4 mutations destabilized the protein. We also obtained the strains described in the Kedziora et al. paper and found low steady state Rif1 protein levels in their strains as well (*Figure 1—figure supplement 2A,F,G*). The low levels of Rif1 protein likely account for the long telomeres seen in the Kedziora et al. study. From our results and the results from *Mattarocci et al., 2014*, we conclude that PP1 binding to Rif1 regulates origin activation, and our data further indicates this interaction is dispensable for telomere length regulation.

### Multiple steps regulate dormant/telomeric origin firing

Our results suggest that several different mechanisms can independently regulate telomeric origin firing. The deletion of *RIF1* caused origin activation in a subset of cells; however, many more cells activated telomeric origins in both ΔN-dbf4 overexpression and in *sld3-A bob1* mutants, as indicated by the higher peaks at ARS608, 608.5, and 609 (*Figures 1C*, *2C* and *3B*). Other groups have also found that *rif1Δ* has only a small effect on origin firing (*Hafner et al., 2018*). The Diffley lab showed that, while *sld3-A* mutants activate some origins, there is an additive effect of mutating both *sld3-A* and *bob1*, indicating that independent mechanisms of activation can have an additive effect on origin firing (*Sheu et al., 2016*; *Zegerman and Diffley, 2010*).

In our analysis, the height of the peak at the origin is linked to the number of cells that fired the origin, while the width of the peak indicates how far the fork is able to travel in HU. This is emphasized in the origin firing graphs of *sld3-A bob1* (*Figure 3B*), which have narrower peaks than seen in *WT* cells, since most origins fire but fork movement is limited by low nucleotide availability in HU. The fact that almost all origins fire demonstrates that, in *sld3-A bob1* mutants, origin firing is not limited by availability of replication complex factors.

PP1 interacts with Rif1 to regulate origin firing through dephosphorylation of Mcm4 (*Davé et al., 2014*; *Hiraga et al., 2014*; *Mattarocci et al., 2014*) and possibly other substrates. While all three methods that we used to activate origins: *rif1-pp1bs*, ΔN-Dbf4 overexpression, and *sld3-A bob1* mutations, resulted in increased Mcm4 phosphorylation, the level of Mcm4 phosphorylation did not

correlate with the degree of origin activation. For example, *rif1Δ* caused a higher increase in Mcm4 phosphorylation compared to both ΔN-dbf4 Cdc7 overexpression and *sld3-A bob1* (*Figures 2B* and *3A*); however, the activation of telomeric origins was much greater in both ΔN-dbf4 Cdc7 overexpression and the *sld3-A bob1* mutant (*Figure 2—figure supplement 1D*, *Figure 3C*). This suggests that additional mechanisms, other than Mcm4 phosphorylation, likely also regulate dormant/telomeric origin firing. Our data supports the conclusion (*Sheu et al., 2016*; *Zegerman and Diffley, 2010*) that there are independent pathways that activate origins, and activation of multiple pathways can result in a higher fraction of cells firing dormant origins.

## Rif1 regulates telomeric origin firing when not bound to the telomere

Rif1 was first identified as a protein that bound to yeast telomeres through its interaction with Rap1 (*Hardy et al., 1992*). Yeast telomeres were known to replicate late in S phase, and when an early firing origin was relocated to a telomere, that origin then fired late (*Ferguson et al., 1991*; *Ferguson and Fangman, 1992*). This suggested that the telomere location or telomeric chromatin causes inefficient origin firing. When Rif1 was discovered to regulate origin firing (*Davé et al., 2014*; *Hiraga et al., 2014*; *Mattarocci et al., 2014*; *Peace et al., 2014*), the solution to the late replication of telomeres appeared simple: the high local concentration of Rif1 at the telomere could dephosphorylate Mcm4 (and likely other Pre-RC components) and block local telomeric origin firing. Our data, however, suggests this model is incomplete. We did not find activation of telomeric origins in *rif1-Δ1322*, which lacks the Rap1 binding domain of Rif1. This *rif1-Δ1322* C-terminal truncation has been shown to block interaction of Rif1 with Rap1 (*Hiraga et al., 2014*; *Shi et al., 2013*) and cause loss of Rif1 telomeric localization (*Hafner et al., 2018*; *Hiraga et al., 2018*). The lack of telomeric origin activation in the *rif1-Δ1322* mutant was surprising, as we expected that release of Rif1 from telomeres would release PP1 repression near telomeres. However, in support of our conclusions, the same C-terminal truncation, in their work termed *rif1-ΔC594*, also showed no effect on origin firing (*Hiraga et al., 2018*). In addition, a two amino acid substitution in Rif1 that disrupts the Rif1-Rap1 binding, *rif1$_{RBM}$*, showed some telomeric origin activation but not as significant as *rif1Δ* (*Hafner et al., 2018*). Perhaps there are other functional domains that are deleted in *rif1-Δ1322* that affect origin firing. Mutations in the HOOK region of Rif1 are proposed to disrupt a mechanism for binding of Rif1 to DNA (*Mattarocci et al., 2017*). We found that, while these mutations caused significant telomere elongation, there was no effect on telomeric origin firing (*Figure 4*). Together, the observations that *rif1$_{HOOK}$* and *rif1-Δ1322* showed no increase in Mcm4 phosphorylation or dormant/telomeric origin activation suggest that mechanisms other than telomere localization of Rif1 may be responsible for repression of dormant/telomeric origins.

## Rif1 regulation of origin firing is conserved while its role in telomere length regulation is not

Rif1 protein is conserved across a wide range of phyla and the role of Rif1 in replication timing is conserved from yeasts to *Drosophila* to humans (*Cornacchia et al., 2012*; *Hayano et al., 2012*; *Yamazaki et al., 2012*). In mammals, while the PP1 binding domain is located in the C terminal region of Rif1 rather than the N terminal region, the interaction with PP1 still regulates late replicating regions. New roles for Rif1 have also been identified in the stabilization and processing of double strand breaks (*Buonomo et al., 2009*; *Callen et al., 2013*; *Xu et al., 2010*). Rif1 interacts with 53BP1 and the Shieldin complex to process double strand DNA breaks, which affects the choice between homologous recombination and NHEJ. This choice plays a major role in immunoglobulin class switch recombination, and thus loss of Rif1 has dramatic consequences for immunoglobulin rearrangements (*Callen et al., 2020*; *Chapman et al., 2013*; *Di Virgilio et al., 2013*; *Silverman et al., 2004*).

Notably, while many of these Rif1 functions in homologous recombination and NHEJ are conserved across phyla (*Fontana et al., 2018*; *Mattarocci et al., 2017*) and (reviewed in *Mattarocci et al., 2016*), the role of Rif1 in telomere length regulation is not conserved in mammals (*Buonomo et al., 2009*; *Silverman et al., 2004*; *Xu and Blackburn, 2004*). Rif1 regulates telomere length in a subset of species including *S. pombe* (*Kanoh and Ishikawa, 2001*), where it can interact with the telomere binding protein Taz1, and in *C. glabrata* (*Castaño et al., 2005*), where it interacts with Rap1. This ability to interact with telomere binding proteins may have been acquired in yeasts,

which imparted a new role in telomere length regulation in these species (reviewed in *Mattarocci et al., 2016*). Remarkably, in yeast engineered to have a chromosome with TTAGGG telomere repeat sequence, to which Rap1 does not bind, telomere equilibrium is maintained, and Rif1 deletion has no effect on the length of this telomere (*Brevet et al., 2003*). These data suggest that Rif1 regulation of telomere length is specific to settings where Rif1 interacts with the telomere binding proteins.

## DNA replication and telomere length

We have shown that the function of Rif1 in regulating origin firing is separable from its function in regulating telomere length. Further, PP1 recruitment is not necessary for telomere length regulation. However, while origin firing is not linked to telomere length, there is still significant evidence linking components of the replication machinery to telomere length (*Dionne and Wellinger, 1998*; *Greider, 2016*). Several studies have suggested a specific role of Rif1 in telomere length regulation. Rif1 has been shown to block telomere elongation at clustered telomeres in late S phase, which may be a clue as to how it negatively regulates telomere addition (*Gallardo et al., 2011*). Shi et al. suggest that interactions of Rif1 and Rif2 together with Rap1 form a 'molecular Velcro' that blocks telomerase access to the telomere (*Shi et al., 2013*). Hirano et al. suggest that both Rif1 and Rif2 block Tel1 recruitment to the telomeres (*Hirano et al., 2009*). Recent evidence indicates that Rif2 regulates Tel1 through catalytic modulation of the MRX complex (*Hailemariam et al., 2019*). This suggests a possibility that Rif1, like Rif2, might also have roles in telomere length regulation other than blocking Tel1 telomere localization.

The remarkable ability of distantly related species to maintain telomere length equilibrium argues for conserved core components of the regulatory process. Since yeast utilize defined sequence-specific origins while mammalian cells use a more stochastic mechanism to activate origins, perhaps it is not surprising that origin firing is not a major component of telomere length regulation. It would be hard to imagine how these two different methods of origin activation would both lead to well-regulated telomere length distributions. The identification of a telomerase-specific RPA protein (*Gao et al., 2007*), which binds single-stranded telomeric DNA at the replication fork and interacts with telomerase, offers one alternative mechanism linking replication to telomere elongation. It is intriguing that Rif1, in addition to its role in regulating origin firing, stabilizes stalled forks and regulates fork progression (*Mukherjee et al., 2019*; *Munden et al., 2018*; *Xu et al., 2010*). We currently do not know how Rif1 regulates telomere elongation; however, establishing that there are clear separable functions of Rif1 will allow dissection of the molecular mechanism of its telomere length regulation function.

## Materials and methods

**Key resources table**

| Reagent type (species) or resource | Designation | Source or reference | Identifiers | Additional information |
|---|---|---|---|---|
| Strain, strain background (*Saccharomyces cerevesiae*) | *Saccharomyces cerevisiae,* CVy61 (W303-1a) | DOI: 10.1002/yea.1406 | | |
| Gene (*Saccharomyces cerevesiae*) | *RIF1* | NA | YBR275C | |
| Gene (*Saccharomyces cerevesiae*) | *DBF4* | NA | YDR052C | |
| Gene (*Saccharomyces cerevesiae*) | *CDC7* | NA | YDL017W | |

*Continued on next page*

Continued

| Reagent type (species) or resource | Designation | Source or reference | Identifiers | Additional information |
|---|---|---|---|---|
| Gene (*Saccharomyces cerevesiae*) | *MCM4* | NA | YPR019W | |
| Gene (*Saccharomyces cerevesiae*) | *MCM5/mcm5-bob1* | *gift from John Diffley:* DOI: 10.1038/nature09373 | YLR274W | |
| Gene (*Saccharomyces cerevesiae*) | *SLD3/sld3-A* | *gift from John Diffley:* DOI: 10.1038/nature09373 | YGL113W | |
| Antibody | anti-FLAG (mouse monoclonal) | Sigma-Aldrich | Sigma-Aldrich: F3165; RRID:AB_259529 | (1:1000) |
| Antibody | anti-V5 (mouse monoclonal) | Thermo Fisher Scientific | Thermo: R960-25; RRID:AB_2556564 | (1:2000) |
| Antibody | anti-HA (mouse monoclonal) | Sigma-Aldrich (Roche) | Sigma-Aldrich:11583816001; RRID:AB_514505 | (1:3000) |
| Antibody | anti-MYC (mouse monoclonal) | Santa Cruz Biotechnology | Santa Cruz:sc-40; RRID:AB_627268 | (1:3000) |
| Antibody | anti-PGK1 (mouse monoclonal) | Thermo Fisher Scientific | Thermo:459250; RRID:AB_2532235 | (1:10000) |
| Antibody | anti-Mouse IgG HRP (horse) | Cell Signaling Technology | Cell Signaling:7076; RRID:AB_330924 | (1:10000) |
| Chemical compound, drug | Hydroxyurea | US Biological | US Biological: H9120 | 200 mM |
| Chemical compound, drug | Alpha Factor | Sigma or US Biological | Sigma:T6901; US Biological: Y2016 | 25 ng/ml for *bar1Δ*, 8 µg/ml +6 µg/ml for *BAR1* |
| Chemical compound, drug | Sytox Green | Invitrogen | Invitrogen:S7020 | 0.5 µl/ml (2.5 µM final) |
| Software, algorithm | FlowJo 10.6.1 | https://www.flowjo.com | RRID:SCR_008520 | |
| Software, algorithm | Adobe Illustrator CS6 | https://www.adobe.com/products/illustrator.html | RRID:SCR_010279 | |
| Software, algorithm | GE Healthcare Life Sciences ImageQuant v8.1 | https://www.gelifesciences.com/en/us/shop/protein-analysis/molecular-imaging-for-proteins/imaging-software/imagequant-tl-8-1-p-00110 | RRID:SCR_018374 | |
| Software, algorithm | GraphPad Prism 5 | www.graphpad.com | RRID:SCR_002798 | |
| Software, algorithm | RStudio 1.2.1335 | www.rstudio.com/ | RRID:SCR_000432 | heatmap.2 https://www.rdocumentation.org/packages/gplots/versions/3.0.3/topics/heatmap.2 |

*Continued on next page*

*Continued*

| Reagent type (species) or resource | Designation | Source or reference | Identifiers | Additional information |
|---|---|---|---|---|
| Software, algorithm | SnapGene 5.0.8 | www.snapgene.com | RRID:SCR_015052 | |
| Software, algorithm | CellQuest Pro 5.2.1 | https://www.bdbiosciences.com/documents/15_cellquest_prosoft_analysis.pdf | RRID:SCR_014489 | |
| Software, algorithm | Bedtools Multicov | https://bedtools.readthedocs.io/en/latest/ | RRID:SCR_006646 | DOI: 10.1002/0471250953.bi1112s47 |
| Software, algorithm | Samtools | http://www.htslib.org | RRID:SCR_002105 | |
| Software, algorithm | BWA-MEM | https://github.com/lh3/bwa | RRID:SCR_010910 | |
| Software, algorithm | Picard MarkDuplicates | https://broadinstitute.github.io/picard/ | RRID:SCR_006525 | |
| Strains | All strains | *Supplementary file 1* | | |
| Recombinant DNA reagent | All plasmids | *Supplementary file 1* | | |
| Sequence based reagent | All primers and oligos | *Supplementary file 1* | | |
| Cloning reagents | Cells and restriction enzymes | *Supplementary file 1* | | |

## Yeast culturing and transformation

Yeast transformation and CRISPR/Cas9 modification were carried out as described in *Keener et al., 2019*. In brief, we used 50 µl of logarithmically growing cells, washed and resuspended in 0.1M Lithium acetate (LiAc, Sigma) with DNA to transform. We added 500 µl of 40% PEG (Polyethylene glycol, P4338; Sigma), 0.1M LiAc, and equilibrated at 30°C for 30 min and then performed heat shock step at 42°C. When selecting for a drug resistance marker, we allowed 4 hr of recovery time in yeast extract peptone dextrose (YPD). To make genome alterations by CRISPR/Cas9, we followed the above transformation protocol as described, but added both ~500 ng of plasmid pJH2972 (gift of Haber lab) (*Anand et al., 2017*) containing a gRNA targeting an NGG close to the site of interest and >1 µg of repair template. We used PCR, restriction digests, and Sanger sequencing to validate transformants.

For passaging yeast strains, cells were grown overnight at 30°C in 5 ml YPD to saturation, and then 5 µl of saturated culture was added to 5 ml of fresh YPD for another overnight incubation. This was repeated for the indicated number of days. At each passage, 1.5 ml of saturated culture was collected, and the pellet was frozen. All genomic DNA was prepared simultaneously. Passage number is indicated in the relevant Southerns.

## Molecular cloning

Plasmids were constructed using PCR fragments and restriction cloning or Gibson assembly methods using Gibson Assembly Master Mix (New England Biolabs E5510). Plasmids and assembly templates were designed *in silico* using SnapGene software (GSL Biotech) and then constructed using PCR with Phusion HS II DNA polymerase (Thermo Fisher F549) from genomic DNA, synthetic G-blocks (IDT), and/or plasmids, followed by TA-cloning and/or Gibson assembly. Plasmids were transformed into NEB5α competent cells (NEB C2987H) for Gibson Assembly or TOP10 competent cells (Thermo Fisher K204040) for TA cloning. Plasmids were prepared using QIAprep Miniprep Kit (Qiagen 27106). Constructs were confirmed by restriction digest and Sanger sequencing. Constructs used to generate strains, including the plasmid or genomic DNA template and oligonucleotides or enzymes

used to prepare the homologous repair DNA, are listed in *Supplementary file 1*. All oligonucleotides and synthetic G-blocks were generated from IDT.

## Southern blot analysis

Southern blot analysis was carried out as described in *Keener et al., 2019*. In brief, to isolate genomic DNA (gDNA), we collected 1.5 ml of saturated culture and vortexed in lysis buffer, with 0.5 mm glass beads and phenol chloroform using a Microtube Foam Insert (Scientific Industries 504-0234-00) for 2 min to lyse the cells. The sample was spun in a centrifuge for 10 min at 14k rpm, and then gDNA was precipitated with 95% ethanol. Finally, we washed the gDNA with 70% ethanol, and, after drying the pellet, resuspended the gDNA in 50 µl TE with RNaseA at 37°C or overnight at 4°C. For restriction digestion, we cut 10 µl of gDNA with *XhoI* (NEB R0146) to visualize Y' telomere fragments. We loaded genomic digests and 100 ng of 2-log ladder (NEB N3200) onto a 1% agarose gel and subjected it to electrophoresis in 1XTTE overnight at 49V. The DNA was then vacuum transferred onto Hybond$^+$ Nylon (GE Healthcare RPN303B) in 10XSSC, and blocked in Church buffer at 65°C. A $^{32}$P radiolabelled Y' PCR fragment (oligo sequences in *Supplementary file 1*) or 2-log ladder (NEB N3200L) was added at $10^6$ counts/ml of Y' and $10^4$ counts/ml of 2-log ladder and hybridized overnight. The Southern was washed with 1XSSC + 0.1% SDS and imaged using a Storm 825 phosphorimager (GE Healthcare) usually after overnight exposure and analyzed with ImageQuant software.

## Western blot analysis

Protein lysates were made using trichloroacetic acid (TCA) extraction (*Link and LaBaer, 2011*). 500 µl or 2OD of cells were collected and resuspend in 5 ml of 10% TCA for 30 min. Pellets were washed with 1M HEPES and resuspended in 2XLDS loading buffer (Invitrogen NP0008) supplemented with 100 mM DTT. 0.5 mm glass beads were added and samples were vortexed on Microtube Foam Insert (Scientific Industries 504-0234-00) for 3 min and then boiled for 5 min. Lysates were spun for 10 min at 14k rpm, and 8 µl of the supernatant was resolved by gel electrophoresis along with 4 µl of Trident Pre-stained protein ladder (GeneTex GTX50875). Mcm4-2XFLAG phosphorylation, Dbf4-V5, and Cdc7-HA were resolved on 3–8% Tris-Acetate gels (Invitrogen EA0375) at 150V for 1 hr. Rif1-13XMyc was resolved on 4–12% Bis-Tris gels (Invitrogen NP0322) at 200V for 1 hr. Mcm4-FLAG, Dbf4-V5, and Cdc7-V5 were transferred onto PVDF membranes using the Trans-Blot Turbo Transfer System (Bio-Rad) using the pre-set 10 min high MW program. Rif1-myc gels were transferred onto PVDF membranes by NuPAGE XCell II Blot Module (EI9051) for 90 min at 30V. The membrane was blocked in Odyssey buffer (Li-Cor 927–40000) except when blotting for FLAG, in which case it was blocked in 1XTBS-T with 5% milk (Bio-Rad 170–6404). The αMyc antibody was used at 1:3000 (Santa Cruz 9E10 c-myc). αFLAG was used at 1:1000 (Sigma Aldrich M2 Flag F1804). αV5 was used at 1:2000 (Invitrogen R96025). αHA was used at 1:3000 (MilliporeSigma 12CA5 11583816001). αPgk1 was used at 1:10,000 (Invitrogen 22C5D8). Secondary HRP-conjugated αMouse antibody (Bio-Rad 1706516) was used at 1:10,000 in 5% milk 1XTBST. For Pgk1 we used SuperSignal West Pico PLUS Chemiluminescent Substrate (Thermo 34580), and for all other primary antibodies we used Forte HRP substrate (Millipore WBLUF0100). Membranes were visualized using ImageQuant LAS4000 (GE Healthcare) and analyzed with ImageQuant software.

## Cell cycle synchronization and flow cytometry

MAT-a cells were grown to OD$_{600}$ of 0.35–0.6. Alpha factor (US Biological Y2016) was added for 2 hr at 25 ng/ml for *bar1Δ* cells and 8 µg/ml supplemented with 6 µg/ml more after 1 hr for *BAR1* cells. Cells were collected, washed in YPD, and resuspended in fresh media to a density of OD$_{600}$0.6–0.8. Cells were released into pre-warmed YPD containing 200 mM Hydroxyurea (HU) (US Biological H9120). Cells were collected after 90 min.

For Mcm4-P synchronization experiments, 2OD of cells were collected after alpha factor synchronization (G1 sample) and 90 min in HU (S sample), spun, and frozen in a dry ice/ethanol bath. TCA extraction and western blot analysis was performed as described above.

For copy number sequencing experiments, 6OD of cells were collected after alpha factor synchronization (G1 sample) and 90 min in HU (S sample), spun, and frozen in a dry ice/ethanol bath. Genomic DNA was prepared as described above using phenol chloroform extraction followed by ethanol

precipitation. Genomic DNA was sent to the Johns Hopkins Genetics Resource Core Facility and High Throughput Sequencing Center and libraries were prepared using a KAPA HyperPlus Kit (Roche), and sequenced using Illumina NovaSeq 2 × 50 paired end reads.

Data analysis was done with assistance from Dr. Sarah Wheelan and Anuj Gupta. Bioinformatics pipeline is detailed below.

For flow cytometry analysis, 500 µl of cells were collected in 750 µl of ice-cold water at the indicated time-points. Cells were collected and fixed in 1 ml of ice-cold 70% ethanol overnight at 4°C. The following day, cells were collected, washed in 500 µl of 50 mM sodium citrate buffer, pH7.5 (NaCitrate), and resuspended in 500 µl of 100 µg/ml RNAseA in 50 mM NaCitrate pH7.5 and incubated at 50°C for several hours. Sytox Green (Thermo S7020) was added to a final concentration of 2.5 µM. Cells were stained in the dark at 4°C for at least 30 min to overnight. Cells were sonicated and strained through the filter cap into the FLOW tube (Thermo 877123). Flow cytometry data was collected on a FACSCalibur using CellQuest software (Becton Dickinson). Data analysis was done using FlowJo software.

## Lambda phosphatase experiment

S phase pellets were collected and frozen as in Mcm4-P synchronization experiments above. Pellets were thawed in an ice water bath and resuspended in 140 µl of phosphatase buffer (1X PMP, 1 mM MnCl2, 2 mM PMSF) or phosphatase inhibitor buffer (1X PMP, 1 mM MnCl2, 2 mM PMSF, 1X Phos-Stop MilliporeSigma 4906837001). 250 µl 0.5 mm glass beads were added, and samples were bead beaten at 4°C for 1 min, and then cooled in an ice water bath for 1 min. Samples were centrifuged for 1.5 min in 4°C at 13k rpm. 20 µl of lysate was taken for analysis, and 400U (1 µl) of λ-phosphatase (NEB P0753S) was added to the phosphatase sample. All samples were incubated at 30°C for 30 min. 20 µl of 4X LDS buffer (Invitrogen NP0008) supplemented with 100 mM DTT was added to stop the reaction, and samples were boiled for 5 min. 15 µl of sample was loaded onto a 3–8% Tris-Acetate gel and western blot analysis was performed as described above. Protocol adapted from *Lucena et al., 2017*.

## Copy number seq analysis

Reads were aligned to SacCer3 using BWA-MEM and indexed using Samtools. Duplicates were marked and removed using Picard MarkDuplicates. Counts of sequence read coverage in 250 bp bins were computed using Bedtools Multicov and normalized to the total number of reads in each sample. The normalized S phase read count was divided by the normalized G1 phase read count to compute a BedGraph of the ratio of S to G1 phase read counts (*Source code 1* and *Supplementary file 2*).

Relative copy number scores of each origin were computed by averaging the ratio of S to G1 phase read counts for 1 kb centered at each OriDB confirmed origin (*Supplementary file 3*). *WT* value was subtracted from mutant value, and the resulting score was multiplied by 100. This is termed Relative Fraction of Origins Fired. Because subtelomeric repetitive elements cannot be uniquely mapped to individual telomeres, X and Y' origins were not included in the analysis.

Chromosome VI graphs were generated using GraphPad Prism 5.0, with chromosome coordinate plotted on the X axis and the ratio of S to G1 phase read counts on the left-hand Y axis termed Relative Copy Number (*Supplementary file 2*). Relative Fraction of Origins Fired was plotted on the right Y axis.

The dot plots of Relative Fraction of Origins Fired were generated using GraphPad Prism 5.0 software, with the median and interquartile range plotted on top of each distribution. A one-sided nonparametric Wilcoxon signed rank test with the alternative hypothesis greater than 0 was performed in R to determine if the true median of the distributions was statistically greater than 0.

The relative copy number of each origin was graphed in GraphPad Prism 5.0 as scatter plots with *WT* on the X axis and mutant on the Y axis. As with the previous analyses, these scores were computed by averaging the ratio of S to G1 phase read counts for 1 kb centered at each OriDB confirmed origin. Each value was then multiplied by 1.25 to account for DNA replication progression in HU before being plotted in Prism (*Supplementary file 3*).

Origin efficiency heatmaps of the ratio of S to G1 phase read counts for 10 kb centered at each OriDB confirmed origin were generated using heatmap.2 in R (*Supplementary file 3*).

The relative score for ARS608 was computed by dividing the relative copy number score of ARS608 by ARS607, and then normalizing the result to that of the *WT* control using the same normalization analysis as used in Relative Fraction of Origins Fired as described above.

## Acknowledgements

We acknowledge Dr. David Mohr and the Johns Hopkins Genetics Resource Core Facility and High Throughput Sequencing Center for preparing and sequencing Illumina genomic DNA libraries. We thank Dr. Sarah Wheelan and Anuj Gupta from the SKCCC Experimental and Computational Genomics Core (NIH P30 CA006973) for advice and assistance in bioinformatics analysis of the copy number seq experiments. We would like to thank Drs. John Diffley and Anne Donaldson for strains and Dr. James Haber for the Cas9 plasmid and protocol. We thank Kathryn Carson at the Johns Hopkins Biostatistics Center for statistics consultation. We thank Drs. Thomas Kelly and Brendan Cormack for many helpful discussions about experimental design and data analysis. Finally, we thank Dr. Thomas Kelly, Dr. Brendan Cormack, Margaret Strong, Dr. Rebecca Keener, Samantha Sholes, and Carla Connelly for critical reading of the manuscript. This work was supported by the Bloomberg Distinguished Professorship (to CWG), NSF GRFP DGE-1746891 (to CBS), and NIGMS T32 GM007445 (to the BCMB graduate training program).

## Additional information

### Funding

| Funder | Grant reference number | Author |
|---|---|---|
| National Science Foundation | DGE-1746891 | Calla B Shubin |
| National Institute of General Medical Sciences | BCMB graduate training program | Calla B Shubin |
| Bloomberg Distinguished Professorship | | Carol W Greider |

The funders had no role in study design, data collection and interpretation, or the decision to submit the work for publication.

### Author contributions

Calla B Shubin, Conceptualization, Data curation, Software, Formal analysis, Funding acquisition, Validation, Investigation, Visualization, Methodology, Writing - original draft, Project administration, Writing - review and editing; Carol W Greider, Conceptualization, Resources, Supervision, Funding acquisition, Validation, Methodology, Writing - original draft, Project administration, Writing - review and editing

### Author ORCIDs

Calla B Shubin (ID) https://orcid.org/0000-0002-4618-2722
Carol W Greider (ID) https://orcid.org/0000-0002-5494-8126

### Decision letter and Author response

Decision letter https://doi.org/10.7554/eLife.58066.sa1
Author response https://doi.org/10.7554/eLife.58066.sa2

## Additional files

### Supplementary files

• Source code 1. Copy number seq analysis.

• Supplementary file 1. Strains, oligos, plasmids, and cloning strategies.

• Supplementary file 2. BedGraphs from Copy Number Seq analysis of normalized S to G1 250 bp coverage.

- Supplementary file 3. Sequencing analysis including confirmed OriDB origins, explanation of how relative copy number scores were computed, and matrices used to generate heatmaps.
- Transparent reporting form

## Data availability

Strains, plasmids, and plasmid maps are available upon request. Source code is available in Source code 1. Source data has been deposited with a BioProject ID PRJNA627739 and can be accessed using the link https://www.ncbi.nlm.nih.gov/bioproject/PRJNA627739.

The following dataset was generated:

| Author(s) | Year | Dataset title | Dataset URL | Database and Identifier |
|---|---|---|---|---|
| Shubin CB, Greider CW | 2020 | Copy number sequencing in *S. cerevisiae* | https://www.ncbi.nlm.nih.gov/bioproject/PRJNA627739 | NCBI BioProject, PRJNA627739 |

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
