## [Decision Letter]

**Acceptance summary:**

The relationship between replication of the genome and the maintenance of telomere length has been of interest ever since telomerase was discovered. It is known that Rif1 plays a role in both the control of initiation of DNA replication and telomere length control. How telomere length regulation is linked to DNA replication is not known, but the current paper provides data from multiple approaches suggesting that telomere length regulation by Rif1 does not involve a process of activation of telomere proximal origins of DNA replication.

**Decision letter after peer review:**

Thank you for submitting your article "The Role of Rif1 in telomere length regulation is separable from its role in origin firing" for consideration by *eLife*. Your article has been reviewed by two peer reviewers, including Bruce Stillman as the Reviewing Editor and Reviewer #1, and the evaluation has been overseen by Kevin Struhl as the Senior Editor.

The reviewers have discussed the reviews with one another and the Reviewing Editor has drafted this decision to help you prepare a revised submission that does not require any new experiments.

Summary:

This work provides valuable evidence against the idea that subtelomeric origin activation can substantially increase telomere length, as stipulated by one specific model of telomere length regulation, i.e., telomerase travels with replication forks and telomerase delivery to telomere ends is negatively correlated with the distance between the ends and replication origins. The paper does not address how telomere length is controlled by telomerase association with the DNA replication fork, but does address one aspect of the coupling to DNA replication that is present in the literature. Thus, a revised paper is requested, taking into account the comments below.

Specific Comments:

1) The effect of *rif1-pp1bs* on subtelomeric origin firing is rather modest. The effects of ∆N-dbf4/Cdc7 overexpression and *sld3-A mcm5-bob1* are stronger, but still weaker than the more internal origin used in normal cells. On the other hand, the effects of the *sld3-A mcm5-bob1* mutations is stronger. It is unclear what fraction of the subtelomeric Ori608 was utilized in the cell population with a particular genetic manipulation and this should be discussed. It is also worth noting that Ori608 is only about 20% closer to chromosome ends than the strong and more internal Ori607, hence not necessarily expected to deliver significantly more telomerase even if it is efficiently activated.

2) Further compounding the analysis is the challenge posed by telomere length heterogeneity; if the changes in telomere lengths are mild, the standard telomere Southern may not be sufficiently sensitive to detect them. For example, Figure 2E suggests that ∆N-dbf4/Cdc7 overexpression strain may have slightly longer telomeres, but rigorously testing this will require higher resolution and more quantitative assays. This might be suggested, but it is clear that the dramatic telomere elongation is not observed.

3) Figures 1C and Figure 1—figure supplement 1E. The extent of late origin firing in the rif1 mutants is relatively low compared to the known early origins. There are two possibilities, one is that only in a very limited number of cells are the late origins activated or because of limited nucleotide (due to HU) or limiting origin firing factors, late origins are activated in all cells but DNA replication is limited. These possibilities should be discussed.

4) The authors cite Nourarede et al. as showing that Mcm4 is phosphorylated by Dbf4-Cdc7. This is not correct and indeed this reference does not even mention Mcm4. Nourarede et al. analyzed Mcm2 phosphorylation, which was known before this paper. The correct reference for Dbf4-Cdc7 phospho-regulation of Mcm4 should be Sheu at al., 2006.

5) There actually is some very good evidence in the literature showing that Rif1 blocks telomerase access to clustered telomeres and only short telomeres are able to break through in S-phase (see Gallardo et al., 2011). Perhaps this mechanism could be suggested as an origin independent mechanism for Rif1 regulating telomere length.

6) A report on how Rif1 and Rif2 might suppress the binding of Tel1, a positive telomerase regulator that is not linked to replication could be mentioned (see Hirano et al., 2009).

---

## [Author Response]

Specific Comments:1) The effect of rif1-pp1bs on subtelomeric origin firing is rather modest. The effects of ∆N-dbf4/Cdc7 overexpression and sld3-A mcm5-bob1 are stronger, but still weaker than the more internal origin used in normal cells. On the other hand, the effects of the sld3-A mcm5-bob1 mutations is stronger. It is unclear what fraction of the subtelomeric Ori608 was utilized in the cell population with a particular genetic manipulation and this should be discussed. It is also worth noting that Ori608 is only about 20% closer to chromosome ends than the strong and more internal Ori607, hence not necessarily expected to deliver significantly more telomerase even if it is efficiently activated.

We thank the reviewers for their suggestion to compare the relative degree of activation of Ori608, which we have referred to as “ARS” 608, in the various mutant backgrounds. To address this, we have now computed a relative activation score for ARS608. The ratio of S phase activation of ARS608 relative to ARS607 in the mutants were normalized to that of the WT control to generate the relative ARS608 activation score (see Materials and methods). By normalizing to ARS607, a higher score indicates that a greater fraction of cells utilized ARS608. We found that *sld3-A bob1* and ∆N-dbf4 Cdc7 overexpression showed the highest relative activation scores, 24.36 and 25.83 respectively. ∆N-dbf4 overexpression alone had a score of 14.09, while *rif1-pp1bs* was 10.15 and *rif1∆* was 6.54. In contrast, *rif1-∆1322* and *rif1_HOOK_* had low ARS608 scores of 1.50 and 2.33 respectively, but have long telomeres. Since telomere elongation is significant in *rif1∆* and this mutant has the lowest relative activation score of those that increase telomeric origin firing, we expect that the mutants with higher scores would show significantly long telomeres if origin firing is linked to telomere elongation. We have added this information to the Discussion.

The reviewer is correct that ARS607 and 608 are 17 kb apart; given that telomeres range from around 250-350 bp, a difference of 17 kb is substantial. In most cells, ARS607 fires, while ARS608 does not fire and is passively replicated. If *rif1∆* telomeres are long due to activation of ARS608, we would expect to detect this change in telomere length in the mutants that activate ARS608 even more than *rif1∆*, even though ARS607 is 17 kb away.

2) Further compounding the analysis is the challenge posed by telomere length heterogeneity; if the changes in telomere lengths are mild, the standard telomere Southern may not be sufficiently sensitive to detect them. For example, Figure 2E suggests that ∆N-dbf4/Cdc7 overexpression strain may have slightly longer telomeres, but rigorously testing this will require higher resolution and more quantitative assays. This might be suggested, but it is clear that the dramatic telomere elongation is not observed.

We appreciate the reviewers’ comment that, while detecting the difference between the long *rif1∆* and *WT* telomeres is reproducibly clear, the resolution of a telomere Southern might not detect very subtle telomere length changes. To be sure that our results were reproducible, we examined at least two independent biological replicates, and at least two, and in many cases more, technical replicates for each mutant background. We acknowledge that, if there were a very subtle change, we might not see a change in the equilibrium distribution by Southern blot. Since we are testing whether the long telomeres in *rif1∆* are caused by origin activation, we would expect a visible change in telomere length, as the *rif1∆* telomeres are quite long. The other mutants we tested activate origins even more strongly than *rif1∆.* Thus, if origin activation was a major player in telomere length regulation, these mutants should have at least as large an effect as *rif1∆.* We have added a sentence in the Discussion addressing this point.

3) Figures 1C and Figure 1—figure supplement 1E. The extent of late origin firing in the rif1 mutants is relatively low compared to the known early origins. There are two possibilities, one is that only in a very limited number of cells are the late origins activated or because of limited nucleotide (due to HU) or limiting origin firing factors, late origins are activated in all cells but DNA replication is limited. These possibilities should be discussed.

We appreciate the reviewers’ insight into the interpretation of HU in the origin firing experiments. We acknowledge that HU depletes nucleotide pools. We see in our data that this depletion affects the distance that a replication fork is able to travel, as seen in other studies (Mantiero et al., 2011), which we noted in subsection “Increasing origin activation using mutations in the pre-RC does not increase telomere length”. This effect is most evident in the *sld3-A bob1* mutant, which has peaks at almost every origin, indicating almost all origins have fired, but the peaks are narrow, indicating that the replication forks were not able to travel as far in the presence of HU compared to *WT*. The *sld3-A bob1* mutant also demonstrates that, even when most origins fire, the origins are able to recruit the essential limiting origin firing factors and travel far enough to be resolved as peaks in our assay. The fact that most origins fire in *sld3-A bob1* demonstrates that origin firing is not limited by limiting origin firing factors in this mutant. We have addressed these points in the revised Discussion.

4) The authors cite Nourarede et al. as showing that Mcm4 is phosphorylated by Dbf4-Cdc7. This is not correct and indeed this reference does not even mention Mcm4. Nourarede et al. analyzed Mcm2 phosphorylation, which was known before this paper. The correct reference for Dbf4-Cdc7 phospho-regulation of Mcm4 should be Sheu at al., 2006.

We thank the reviewers for identifying the mistaken citation, and we have added Sheu at al., 2006 as the proper citation.

5) There actually is some very good evidence in the literature showing that Rif1 blocks telomerase access to clustered telomeres and only short telomeres are able to break through in S-phase (see Gallardo et al., 2011). Perhaps this mechanism could be suggested as an origin independent mechanism for Rif1 regulating telomere length.

We appreciate the reviewers’ thinking about a very important question- How does Rif1 regulate telomere elongation? We now address the literature on the role of Rif1 in clustered telomeres in the Discussion.

6) A report on how Rif1 and Rif2 might suppress the binding of Tel1, a positive telomerase regulator that is not linked to replication could be mentioned (see Hirano et al., Mol. Cell 33, 312).

We agree with the reviewers that the interaction of Rif1 and Rif2 with Tel1 in telomere length regulation may provide insight into the role of Rif1 in telomere regulation. The Hirano et al. paper discusses a potential role for both Rif1 and Rif2 in Tel1 localization to the telomere. Recent evidence indicates that Rif2 regulates Tel1 through catalytic modulation of the MRX complex (Hailemariam et al., 2019). We now discuss the possible role of Tel1 recruitment and note that Rif1, like Rif2, might also have roles other than Tel1 recruitment (subsection “DNA replication and telomere length”).